# The Biological Assessment of Shikonin and β,β-dimethylacrylshikonin Using a Cellular Myxofibrosarcoma Tumor Heterogeneity Model

**DOI:** 10.3390/ijms242115910

**Published:** 2023-11-02

**Authors:** Birgit Lohberger, Heike Kaltenegger, Nicole Eck, Dietmar Glänzer, Andreas Leithner, Nadine Kretschmer

**Affiliations:** 1Department of Orthopedics and Trauma, Medical University of Graz, 8036 Graz, Austria; heike.kaltenegger@medunigraz.at (H.K.); nicole.eck@medunigraz.at (N.E.); dietmar.glaenzer@medunigraz.at (D.G.); andreas.leithner@medunigraz.at (A.L.); 2Institute of Pharmaceutical Sciences, Pharmacognosy, University of Graz, 8010 Graz, Austria

**Keywords:** myxofibrosarcoma, shikonin, β,β-dimethylacrylshikonin, apoptosis, MAPK signaling, tumor heterogeneity, DNA damage response

## Abstract

Myxofibrosarcoma (MFS) is a subtype of soft tissue sarcoma of connective tissue, which is characterized by large intra-tumor heterogeneity. Therapy includes surgical resection. Additional chemotherapy is of limited effect. In this study, we demonstrated the potent anticancer activity of shikonin derivatives in our MFS cellular model of tumor heterogeneity for developing a new therapeutic approach. The impact of shikonin and β,β-dimethylacrylshikonin (DMAS) on viability, apoptotic induction, MAPK phosphorylation, and DNA damage response were analyzed by means of two human MFS cell lines, MUG-Myx2a and MUG-Myx2b, derived from a singular tumor tissue specimen. MFS cells showed a dose-dependent inhibition of cell viability and a significant induction of apoptosis. Treatment with shikonin derivatives caused an inhibition of pSTAT3 and an increase in pAKT, pERK, pJNK, and pp38. DMAS and shikonin inhibited the activation of the two master upstream regulators of the DNA damage response, ATR and ATM. MUG-Myx2b, which contains an additional *PTEN* mutation, was more sensitive in some targets. These data demonstrate the significant antitumorigenic effect of shikonin derivatives in MFS and highlight the importance of intra-tumor heterogeneity in treatment planning.

## 1. Introduction

Myxofibrosarcoma (MFS) represents a heterogeneous subtype of malignant soft tissue sarcoma that arises from the connective tissues, specifically the fibrous and fatty tissues. It is characterized by a mixture of myxoid and fibrous components. In accordance with the 2020 World Health Organization Classification of Soft Tissue Tumors, it falls within the category of fibroblastic/myofibroblastic tumors [1]. MFS most commonly affects the extremities, particularly in older adults with a slight male predominance, but it can also occur in other areas of the body. The tumor grows locally aggressive, has the potential to metastasize, and is clinically characterized by a high risk of local recurrence related to its infiltrative growth pattern. MFS has been categorized into three grades, which are determined by assessing factors like cellularity, nuclear pleomorphism, and proliferative activity [2]. In contrast to low-grade MFS, intermediate- and high-grade MFS tumors exhibit significantly complex karyotypes, featuring multiple copy number alterations. As a result of their more aggressive behavior, these higher-grade MFS tumors are associated with a less favorable prognosis. The treatment options in advanced disease are limited. Wide resection stands as the established treatment protocol for localized disease. In surgical practice, the choice of procedure within a tumor board for each specific patient is based on various factors such as the size, location, and stage of the tumor, its interaction with adjacent neurovascular and bony structures, and the functional and cosmetic requirements of the patient. Rhee et al. recommended minimal resection margins of at least 2 cm for resection of MFS [3]. Radiation therapy can be employed as neoadjuvant or adjuvant treatment approaches to enhance local tumor control. While the role of radiation therapy in the treatment of high-grade MFS is controversial, several retrospective studies have suggested that combining radiation therapy with surgery is linked to a reduced risk of local recurrence [4,5]. Adjuvant chemotherapy is applied depending on the stage and grade with limited effect [6]. Like other sarcoma subtypes, the anthracycline doxorubicin (with or without ifosfamide) is the first-line treatment for advanced MFS. 

To explore new treatment options, we utilized plants derived from traditional Chinese medicine, which offer a wealth of bioactive compounds. Shikonin is a natural compound derived from the roots of medicinal plants like *Lithospermum erythrorhizon* Siebold et Zucc or *Onosma paniculata Bur. et Franch*. It possesses various biological activities and has been traditionally used in herbal medicine for its anti-inflammatory, antimicrobial, antioxidant, and anticancer properties. Shikonin has attracted attention in scientific research due to its potential therapeutic applications in various diseases, including cancer, inflammation, and skin disorders [7,8]. Shikonin, an extensively researched naphthoquinone derivative, has exhibited remarkable anti-cancer activity across diverse types of cancer cells. It is known to exert its effects on cancer cells through multiple mechanisms, including the inhibition of cell proliferation, induction of apoptosis, and reduction of cell migration and invasion potential [9,10,11]. These actions are mediated through the modulation of various molecular signal transduction pathways [12,13,14]. β,β-Dimethylacrylshikonin (DMAS) is a derivative of shikonin, which is modified by the addition of two methyl groups and an acryl group. This compound exhibits biological activities similar to those of shikonin, including anti-inflammatory, antimicrobial, antioxidant, and anticancer effects. DMAS has been investigated for its potential therapeutic applications and has unique properties compared to shikonin itself [15].

Tumor heterogeneity poses significant challenges in cancer diagnosis, prognosis, and treatment. It can contribute to differences in tumor behavior, response to therapies, and the development of drug resistance. Understanding and characterizing tumor heterogeneity is crucial for personalized medicine approaches, as it can help identify specific subpopulations of cells within the tumor that may require targeted treatments. The MUG-Myx2a and MUG-Myx2b cell lines, derived from a single MFS tumor sample, offer us a distinctive cellular model of tumor heterogeneity [16]. MUG-Myx2a exhibited increased proliferation activity, accelerated migration, and enhanced tumorigenicity. Next-generation sequencing mutation analysis identified mutations in the *FGFR3, KIT, KDR*, and *TP53* genes that corresponded to these findings. Furthermore, the MUG-Myx2a cell lines presented an additional mutation in the *PTEN* gene. 

Despite the existence of published data regarding the effects of shikonin derivatives on different tumor types, there is currently no available information regarding the potential use of shikonin or DMAS for treating MFS. In our present study, we utilized our tumor heterogeneity model to investigate the impact of shikonin and its derivative DMAS on various cellular mechanisms, including apoptosis induction, the regulation of mitogen-activated protein kinases (MAPKs) and signal transducer and activator of transcription 3 (STAT3), as well as DNA damage response.

## 2. Results

### 2.1. Effects of DMAS and Shikonin on Cell Viability and Proliferation 

DMAS is obtained when a β,β’-dimethylacryl group is added to shikonin (Figure 1A). The IC_50_ concentrations were determined by treating both cell lines with increasing amounts of shikonin and DMAS, followed by the determination of the dose–response curves. In both MUG-Myx2a and MUG-Myx2b, shikonin showed significantly higher efficacy on cell survival after 24 h (Figure 1B). The following IC_50_ values were determined: DMAS: 1.38 µM for MUG-Myx2a and 1.55 µM for MUG-Myx2b; shikonin: 0.69 µM for MUG-Myx2a and 0.78 µM for MUG-Myx2b. DMSO in the used concentrations had no effect on cell viability. Within the range of these IC_50_ values, there was a highly significant difference in the efficacy of the two derivatives. To ensure quantitative comparability, the respective IC_50_ value und a concentration below it and a concentration above it were used for all subsequent experiments. This standardized metric allowed direct quantitative comparison of the efficacy of both derivatives. For all protein analyses, the treatment conditions consisted of 0 µM (control), 0.5 µM, 1.5 µM, and 3 µM DMAS, as well as 0.25 µM, 0.5 µM, and 1 µM shikonin. For gene expression analyses, the treatment conditions included 1.5 µM DMAS or 0.5 µM shikonin. Gene expression analyses of both proliferation markers *cMyc* and *survivin* revealed a highly significant reduction upon treatment with both derivatives (Figure 1C; significances were represented with stars). In terms of the intra-tumoral heterogeneity, it was evident that the expression of *cMyc* was less influenced by MUG-Myx2b (in this case the significances were represented with rhombuses). 

### 2.2. Caspase Activity and PARP Cleavage as Hallmarks of Apoptosis 

To investigate the induction of apoptosis in the MFS heterogeneity model, whole cell lysates for Western blot analysis were extracted 24 h after treatment of cells with 0 µM (control), 0.5 µM, 1.5 µM, and 3 µM DMAS, as well as 0.25 µM, 0.5 µM, and 1 µM shikonin. Fold changes normalized to untreated controls (Δ ratio; mean ± SD of *n* = 3) were presented. Higher concentrations of both derivates revealed a clear dose-dependent cleavage of caspase-8, -9, -3 and PARP (Figure 2A). Likewise, the apoptotic antagonist Noxa and the DNA damage marker γH2AX were upregulated. All of this clearly points to an induction of apoptosis. Although the progression of both cell lines was basically similar, minor differences in sensitivity could be observed. More clearly, these differences could be shown in the gene expression analysis (Figure 2B). After treatment with the respective IC_50_ concentrations for 24 h, the RNA was isolated, and gene expression of the pro-apoptotic markers *Bak, Bax, Bim*, the anti-apoptotic gene *Bcl-2*, and the antagonists *Puma* and *Noxa* were analyzed. Untreated cells were used as control (ratio = 1) (mean ± SEM; *n* = 6, measured in triplicate). Relative gene expression of *Bak, Bim, Puma*, and *Noxa* was highly significantly upregulated after treatment, whereas the anti-apoptotic gene *Bcl-2* was, as expected, downregulated. *Bak*, *Bcl-2*, and *Noxa* were significantly differentially expressed by intra-tumor heterogeneity. Thus, the results of the gene expression analyses supported the protein data and indicated an apoptotic induction.

### 2.3. The Expression of Death Receptors Was Influenced by Shikonin Derivatives

Death receptor gene expression was analyzed using RT-PCR. Treatment of MUG-Myx2a/2b cells with the respective IC_50_ concentrations of DMAS/shikonin derivatives for 24 h resulted in a highly significant increase in DR4 (*TNFRSF10A*) and DR5 (*TNFRSF10B*) expression (Figure 3A). In contrast, the expression of the metastatic marker matrix metalloproteases 2 (*MMP2*) showed decreasing trends. Here again, a different degree of regulation between the two cell lines was observed (Figure 3B). Gene expression analysis revealed a significant downregulation of the metastatic marker *MMP2* after treatment with DMAS or shikonin for 24 h (Figure 3B).

### 2.4. Modification of the Phosphorylation Pattern by DMAS and Shikonin

To assess how DMAS and shikonin influence mitogen-activated protein kinases (MAPKs) phosphorylation levels, we collected whole cell lysates from MUG-Myx2a and MUG-Myx2b cells one hour after they were exposed to different concentrations of DMAS (0.5, 1.5, and 3 µM) or shikonin (0.25, 0.5, and 1 µM). These lysates were then prepared for Western blot analysis. As the concentrations of DMAS and shikonin increased, we observed a gradual reduction in STAT3 phosphorylation at the protein level, indicating a dose-dependent effect (Figure 4A). Conversely, we noticed a notable increase in the phosphorylation of the serine/threonine kinase AKT, particularly at the higher concentrations. To elucidate the mechanism responsible for inducing apoptosis, we conducted protein analysis for ERK, JNK, and p38, which are key components of the MAPK pathway. In treated cells, we observed an elevated level of phosphorylated JNK (pJNK) and p38 (pp38) compared to untreated control cells. One representative blot out of three is shown, and β-actin was used as loading control. ∆ represents the ratio of phosphorylated to unphosphorylated MAPKs (mean ± SD; *n* = 3). In this case, both cell lines reacted very similarly.

Figure 4B displays the results of the gene expression analysis for the downstream targets of STAT3, namely, *SOCS3* and *Sox9*. Due to the substantial decrease in STAT3 phosphorylation, the MUG-Myx2b cell line exhibited a highly significant reduction in the expression of *SOCS3* and *Sox9*. In contrast, the response observed in the MUG-Myx2a cell line was different. Untreated cells were used as control (ratio = 1) (mean ± SEM; *n* = 6). 

One potential explanation for the divergent behavior of the two cell lines could be attributed to the *PTEN* mutation that is present in MUG-Myx2b [16]. RNA sequencing data identified mutations in the *FGFR3, KIT, KDR*, and *TP53* genes that corresponded between the two cell lines. However, the MUG-Myx2a cell lines exhibited an additional mutation in the *PTEN* gene. In comparison to the untreated controls, the samples treated with DMAS or shikonin displayed a significant reduction in the gene expression of *MDM2*, *PTEN*, and *p53*. Furthermore, it is important to note that for all three markers, MUG-Myx2a and MUG-Myx2b exhibited distinct differences from each other (Figure 5).

### 2.5. Effect of DMAS and Shikonin on DNA Damage Response

DNA damage response (DDR) is a highly conserved genome surveillance mechanism that preserves cell viability in the presence of therapeutic drugs. Whole-cell proteins were isolated according to the same treatment modalities as in the previous experiments and subsequently, these cell lysates were readied for analysis via the Western blotting technique. Fold changes normalized to untreated controls (Δ ratio; mean ± SD of *n* = 3) were presented (Figure 6).

As the concentrations of shikonin derivatives increased, the phosphorylation of the two master upstream regulators involved in DDR signaling, namely ataxia telangiectasia mutated (ATM) and RAD3-related (ATR), was suppressed. These effects, which were induced by both shikonin and DMAS treatment, were more pronounced in MUG-Myx2a cells. The expression of MSH3 und XPC was also decreased. The phosphorylation patterns of seronin/threonine checkpoint kinases Chk1/2 exhibited significant differences between the two distinct cell lines. The shikonin derivatives induced a dose-dependent augmentation of phosphorylation in the MUG-Myx2a cell line, whereas in the MUG-Myx2b cells, a decrease in phosphorylation was observed.

## 3. Discussion

Given the resistance of MFS to conventional chemotherapy and radiotherapy, the exploration of novel groups of substances and their underlying cellular mechanisms holds paramount significance. Adjuvant chemotherapy consists, as in most sarcoma entities, of first-line therapy with the anthracycline doxorubicin. However, it is not well suited as a comparative substance in cell culture, because it is known that authentic doxorubicin is converted in cell culture media to a chemically distinct form resulting in varying chemosensitivity and activity. This transformation results in a significant loss of lethality in vitro while retaining antiproliferative activity [17].

The active ingredients of traditional Chinese medicine offer a variety of exciting possibilities. The roots of *Lithospermum erythrorhizon* have been reported to exhibit notable anti-cancer effects. Shikonin, a primary active ingredient, emerges as a highly compelling target molecule, demonstrating a wide range of potential applications and a realistic possibility for clinical utilization [7]. Moreover, the diversity and heterogeneity observed in tumors present a significant challenge in the field of cancer treatment. 

For cell culture experiments, we used MUG-Myx2a and MUG-Myx2b, two well-established human MFS cell lines, derived from a singular tumor tissue specimen, providing a valuable cellular model for studying the intra-tumoral heterogeneity of this cancer type [16]. To assess the cytotoxic effects of shikonin and DMAS on human MFS cells, the CellTiter-Glo^®^ assay was employed. This assay quantifies the number of metabolically active cells by measuring ATP levels. Both cell lines exhibited a dose-dependent reduction in cell viability, with IC_50_ values notably lower than those reported for melanoma cells, embryonic kidney cells [18], and chondrosarcoma cells [13]. This observation suggests that human MFS cells could potentially be more sensitive to shikonin and its derivatives compared to other types of tumors. Indeed, apoptotic induction, the modulation of death receptor expression, regulation of MAPK phosphorylation, and the DNA damage response are pivotal cellular mechanisms that significantly contribute to defining the anti-cancer activity of both shikonin and DMAS. The goal in eliminating cancer cells through non-surgical methods is achieved by triggering apoptosis. Our findings demonstrated that shikonin derivatives induced apoptosis in a concentration-dependent manner, ranging from 0.5 to 3 µM for DMAS and 0.25 to 1 µM for shikonin, within MFS cells. The activation of caspases serves as the initiator for programmed cell death, inducing cell membrane swelling, cell contraction, chromatin condensation, and DNA degradation [19]. Our findings revealed a dosage-dependent increase in the cleavage of caspases -8, -9, and -3, along with PARP due to exposure to DMAS and shikonin. In general, apoptotic induction by shikonin has already been shown in other tumor entities [13,20,21]. The intrinsic pathway involves activation of caspase-9, whereas the extrinsic pathway necessitates activation of caspase-8. Our study demonstrated a notable concentration-dependent rise in caspase-8 and caspase-3 activity in MFS cells, suggesting that shikonin derivatives induced apoptosis in our cell system through the extrinsic pathway. Caspase-3 and -7, known as executioner caspases, play a crucial role in the apoptosis process by initiating PARP cleavage, a recognized indicator of apoptosis. In our tumor heterogeneity cell system, we also observed the cleavage of PARP during the apoptosis process in both cell lines. In chondrosarcoma cells, comparable effects were demonstrated following treatment only from a concentration of over 2.5 µM shikonin [13].

*Survivin* (*BIRC5*) is an evolutionarily conserved eukaryotic protein that is essential for cell division and plays a crucial role in the progression of apoptosis. As a member of the apoptosis inhibitor family, *survivin* functions by inhibiting caspase activation, ultimately leading to the negative regulation of apoptosis [22]. Given the alignment of concentrations with the calculated IC_50_ values, the notable and significant downregulation of survivin induced by shikonin and DMAS emerges as a promising mechanism for triggering apoptosis. MUG-Myx2b was significantly more sensitive to DMAS than the MUG-Myx2a cell line. The regulation of this pathway is orchestrated by the *Bcl-2* family of proteins, encompassing both pro-apoptotic and pro-survival members. Their intricate interplay delicately balances the cellular choice between survival and programmed cell death [23]. The pro-apoptotic markers *Bak* and *Bim* exhibited increased expression in myxofibrosarcoma cells, while the anti-apoptotic *Bcl-2* showed a significantly notable decrease in expression. The two antagonists *Puma* and *Noxa* were significantly upregulated by DMAS and shikonin. As demonstrated in our earlier study on melanoma cells, escalating concentrations of shikonin derivatives resulted in a dose-dependent augmentation of *Noxa* expression within our cellular model. The heightened responsiveness of MUG-Myx2b cells was likewise noted for *Bak, Bcl-2*, and *Noxa*.

The activation of DNA double-strand breaks (DSB) can also precipitate cell death caused by shikonin derivatives [24]. This is evident from the increased levels of phosphorylated histone variant γH2AX, serving as a biomarker for DSB. Analyzing the protein phosphorylation of this biomarker for DNA damage, we observed a concentration-dependent rise in its expression for both MFS cell lines. Consistent with apoptotic induction, the death receptors *DR4* and *DR5* were also significantly increased in expression. MMPs play a pivotal role in the metastasis process, owing to their biological functions. These encompass the degradation of extracellular matrix components and interaction with growth factors like cytokines and chemokines [25]. DMAS and shikonin notably reduced the gene expression of *MMP2*, whereas the MUG-Myx2b cells responded significantly more sensitively.

The MAPKs represent a group of serine/threonine kinases responsible for transmitting signals from the cell membrane to the nucleus. They serve as key components in diverse biological processes, including cell proliferation, differentiation, apoptosis, immune responses, and stress responses. Lee et al. documented that treatment with shikonin-induced apoptosis in melanoma cells through activation of the MAPK pathway [26]. Given this context, we conducted protein phosphorylation analyses to ascertain if the apoptosis induced by shikonin in MFS cells is orchestrated through MAPK signaling pathways. In particular, we found a notable decrease in STAT3 phosphorylation following treatment especially with DMAS and a little less also with shikonin. These inhibitory effects have already been demonstrated in chondrosarcoma cells in a very similar way [13]. As phosphorylation is a very fast effect, the 1 h time point was chosen. Conversely, there was an observed elevation in the phosphorylation level of the serine/threonine kinase AKT. Previous studies have established the significance of the AKT pathway in the induction of apoptosis by shikonin across various cancer types [13,27]. In the phosphorylation of MAPKs ERK/JNK/p38, MFS cells exhibited a lesser increase compared to chondrosarcoma cells. Following DMAS/shikonin treatment, in MUG-Myx2a cells there was a significant upregulation of the suppressor of cytokine signaling 3 (*SOCS3*), a known negative regulator of the JAK/STAT signaling pathway, whereas the MUG-Myx2b cells showed a highly significant decreased expression.

Because of the additional *PTEN* mutation of MUG-Myx2a, we investigated the *PTEN-MDM2-p53* tumor suppressor oncoprotein network, which regulates cell growth and viability [28]. Both DMAS and shikonin decreased the gene expression of *PTEN, MDM2*, and *p53*. Genetic variations in these genes could potentially disrupt the cell cycle and promote tumor development.

The DDR is a highly conserved genome surveillance mechanism crucial for maintaining cell viability, especially in the presence of chemotherapeutic drugs. Shikonin was identified as an inhibitor that strongly suppressed DDR in pancreas carcinoma and colorectal cancer cells [29]. In MFS cells, DMAS and shikonin inhibited the activation of ATR and to a lesser degree ATM, two master upstream regulators of the DDR. In addition, the DNA repair markers MSH3 and XPC were downregulated after DMAS treatment. Interestingly, DMAS/shikonin treatment significantly increased phosphorylation of Chk1 in MUG-Myx2a cells, whereas MUG-Myx2b showed a reduction. These new findings establish shikonin derivatives as pan DDR inhibitors and highlight ATM as a pivotal factor in determining the chemotherapy-enhancing effect. 

## 4. Materials and Methods

### 4.1. Cell Culture

MUG-Myx2a and MUG-Myx2b are well-established cell lines derived from a 94-year-old female patient with G3 stage MFS, providing a valuable cellular model for studying the intra-tumoral heterogeneity of this cancer type [13]. Immunohistochemical analysis of the patient’s tumor revealed limited smooth muscle actin (SMA) positivity, while Caldesmon and S100 (Dako, Glostrup, Denmark), CD34 (Neomarkers, Fremont, CA, USA), Desmin, EMA, and Pan-CK (all Ventana Medical Systems, Tucson, AZ, USA) were all negative. 

Following the surgical removal, the finely minced tissue underwent an overnight enzymatic digestion at 37 °C using 2 mg/mL of collagenase B (Roche Diagnostics, Mannheim, Germany). After centrifugation at 1400× *g* rpm for 5 min, two distinct fractions were obtained. Firstly, the cell pellet was subjected to two washes with PBS and then plated in Dulbecco’s-modified Eagle’s medium (DMEM-F12; Gibco™, Thermo Fisher Scientific, Waltham, MA, USA). This medium contained 10% fetal bovine serum (FBS), 1% L-glutamine, 100 units/mL penicillin, 100 µg/mL streptomycin, and 0.25 µg amphotericin B (all Thermo Fisher Scientific). The sub-clone MUG-Myx2b was subsequently cultured from this fraction. Secondly, the viscous colloidal supernatant was collected and cultured as the sub-clone MUG-Myx2a using the aforementioned culture medium. Over a span of 24 months, the cells adopted an adherent monolayer growth pattern and underwent more than 70 passages. The original tissue and the derived cell lines exhibited identical short tandem repeat (STR) profiles for various genetic markers measured with the Power Plex^®^ 16 System (Promega, Vienna, Austria). The resulting data were processed and evaluated using ABI Genemapper 4.0 (Applied Biosystems Inc., Foster City, CA, USA). The MFS cell lines were maintained at 37 °C in a humidified atmosphere with 5% CO_2_. For dose-response analysis and protein and RNA isolation, the cells were treated for a 24 h incubation period. To capture phosphorylation events, proteins related to the STAT3, AKT, and MAPK pathways were isolated after only 1 h of treatment, considering the rapid phosphorylation process.

### 4.2. Viability Assays 

Shikonin (# PHL89791) and DMAS (# SML3463) were purchased from Sigma Aldrich (St. Louis, MI, USA) in a stock solution of 10 mM/DMSO. For viability assays, a total of 5 × 10^3^ MUG-Myx2a/2b cells per well were plated onto white 96-well plates. The cells were divided into control groups and treatment groups, where they were exposed to shikonin or DMAS at various concentrations ranging from 0.1 to 25 µM. The dose–response curves were determined using the CellTiter-Glo^®^ Luminescence Assay from Promega, following the manufacturer’s instructions, after a 24 h incubation period. The absorbance values were measured using the Lumistar^®^ microplate luminometer (BMG Labtech, Ortenberg, Germany). The assay was performed in biological quadruplicate (*n* = 6), and the untreated culture media was used as a reference for background values. The IC_50_ values, representing the concentration at which 50% inhibition of cell viability occurred, were calculated using SigmaPlot 14.5 software from Systat Software Inc. (San Jose, CA, USA). The calculations were based on the four-parameter logistic curve. 

### 4.3. Western Blot Analysis 

After subjecting the cells to treatment with different concentrations of DMAS (0.5 µM, 1.5 µM, and 3.0 µM) or shikonin (0.25 µM, 0.5 µM, and 1.0 µM), the cells were incubated for 60 min to determine phosphorylation levels. For the investigation of apoptotic induction and DNA damage, the cells were incubated with the respective concentrations of DMAS or shikonin for 24 h. Following the incubation period, whole cell protein extracts were prepared using lysis buffer, specifically RIPA buffer (Cell Signaling Technology, Danvers, MA, USA), supplemented with a cocktail of protease and phosphatase inhibitors obtained from Sigma Aldrich. This lysis buffer and inhibitor combination ensures the preservation of protein integrity and prevents unwanted enzymatic activity during the extraction process. The protein samples were separated by sodium dodecyl sulfate-polyacrylamide gel electrophoresis (SDS-PAGE) and transferred onto Amersham™ Protran™ Premium 0.45 µM nitrocellulose membranes (GE healthcare Life science, Little Chalfont, UK). The protein concentration in the samples was determined using the Pierce BCA Protein Assay Kit (Thermo Fisher Scientific), following the manufacturer’s instructions. Primary antibodies against various proteins were used for immunoblotting. These included cleaved-caspase-8, -9, and -3, cleaved-PARP, Noxa, phosphorylated histone H2AX (γH2AX), phospho-AKT^Ser473^, AKT, phospho-STAT3^Tyr705^, STAT3, phospho-ERK^Thr202/Tyr204^, ERK, phospho-JNK^Thr183/Tyr185^, JNK, phospho-p38^Thr180/Tyr182^, and p38 as well as the DNA damage key proteins pATR, pATM, MSH3, XPC, and pChK1/2 (all Cell Signaling Technology). The antibody for the loading control, β-actin, was obtained from Santa Cruz (Santa Cruz Biotechnology, Santa Cruz, CA, USA). Detailed information on antibody dilutions and incubation times can be found in Table 1. The blots were developed using a horseradish peroxidase-conjugated secondary antibody (Dako, Jena, Germany) incubated at room temperature for 1 h. The Amersham™ ECL™ prime Western blotting detection reagent (GE Healthcare) was used according to the manufacturer’s protocol for chemiluminescent signal detection. The ChemiDoc Touch Imaging System (BioRad Laboratories Inc., Hercules, CA, USA) was utilized to capture the chemiluminescence signals, and the images were processed using the ImageLab 5.2 Software, also from BioRad Laboratories Inc.

### 4.4. Reverse Transcription Polymerase Chain Reaction (RT-PCR)

Total RNA was isolated 24 h after treatment with 1.5 µM DMAS or 0.5 µM shikonin using the RNeasy Mini Kit and DNase-I treatment according to the manufacturer’s manual (Qiagen, Hilden, Germany). Two micrograms of RNA was reverse transcribed with the iScript-cDNA Synthesis Kit (BioRad Laboratories Inc.) using a blend of oligo(dT) and hexamer random primers. Amplification was performed with the SsoAdvanced Universal SYBR Green Supermix (Bio-Rad Laboratories Inc.) using technical triplicates and measured by the CFX96 Touch (BioRad Laboratories Inc.). The following QuantiTect primer assays (Qiagen) were used for real-time RT-PCR: the proliferation markers *cMyc* (QT00062069), *survivin* (*BIRC5*; QT01679664); the apoptotic key players *Bak* (QT00228508), *Bim* (*BCL2L11*; QT00079037), *Bcl-2* (QT00025011), *Puma* (*BBC3*; QT00082859), and *Noxa (PMAIP1*; QT01006138); the death receptors *DR4* (*TNFRSF10A*; QT00065723) and *DR5* (*TNFRSF10B*; QT00082768); *MMP2* (QT00088396); the MAPK downstream targets *SOCS3* (QT00244580) and *Sox9* (QT00001498); and the *PTEN* associated genes *MDM2* (QT00056378), *PTEN* (QT00096933), and *p53 (TP53*; QT00060235). Results were analyzed using the CFX manager for CFX Real-Time PCR Instruments (Bio-Rad Laboratories Inc., version 3.1) software, and quantification cycle values (C_t_) were exported for statistical analysis. Results with Ct values greater than 32 were excluded from analysis. Relative quantification of expression levels was obtained by the ∆∆Ct method based on the geometric mean of the internal controls ribosomal protein, large, P0 (RPL; QT00075012)) and TATA box binding protein (TBP; QT00000721), respectively. Expression level (C_t_) of the target gene was normalized to the reference genes (ΔC_t_), and the ΔC_t_ of the test sample was normalized to the ΔC_t_ of the control (ΔΔC_t_). Finally, the expression ratio was calculated with the 2^−ΔΔCt^ method. 

### 4.5. Statistical Analysis 

Statistical analyses were performed using IBM SPSS Statistic 29.0.0.0 (241) (New York, NY, USA), and graphical representation were performed using the SigmaPlot 14.5 software (SYSTAT, Palo Alto, CA, USA). Data were tested for normality with the Kolmogorov–Smirnov test. Since data distribution in all samples significantly deviated from normal distribution, statistical significance of the observed differences was tested with non-parametric tests. Single comparisons were tested using Mann–Whitney U test. Multiple comparisons were tested with Kruskal–Wallis H test, followed by pairwise analysis with Bonferroni correction. *p*-values were considered statistically significant if they were less than 0.05*^/#^, 0.01**^/##^, or 0.001***^/###^, indicating the level of significance.

## 5. Conclusions

Our findings show the extensive anti-tumor efficacy of shikonin and its derivative DMAS, in MFS cells. These compounds significantly impacted cell viability, induced apoptosis through caspase cleavage and regulation of death receptors, and impacted MAPK phosphorylation. Moreover, our study identified shikonin derivatives as potent pan DNA damage response (DDR) inhibitors, emphasizing the crucial role of ATM in this context. 

## Figures and Tables

**Figure 1 ijms-24-15910-f001:**
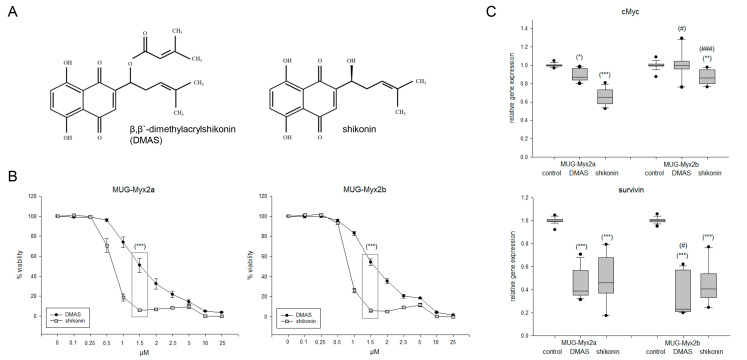
Effects of DMAS and shikonin on cell viability and proliferation. (**A**) Chemical structures of β,β-dimethylacrylshikonin (DMAS) and shikonin. (**B**) Dose–response relationship showing the dose-dependent reduction in cell growth, where shikonin showed a highly significant, more efficient effect than DMAS at a concentration of 1.5 µM. (**C**) Relative gene expression of the proliferation markers *cMyc* and *survivin* 24 h after treatment with the respective IC_50_ concentrations of DMAS/shikonin (mean ± SD; *n* = 6; measured in triplicate). Untreated cells were used as controls (ratio = 1). Statistical significances are defined as follows: *^/#^
*p* < 0.05; ** *p* < 0.01; ***^/###^
*p* < 0.001 (controls vs. DMAS/shikonin treated cells were represented with stars; significances between both cell lines were represented with rhombuses).

**Figure 2 ijms-24-15910-f002:**
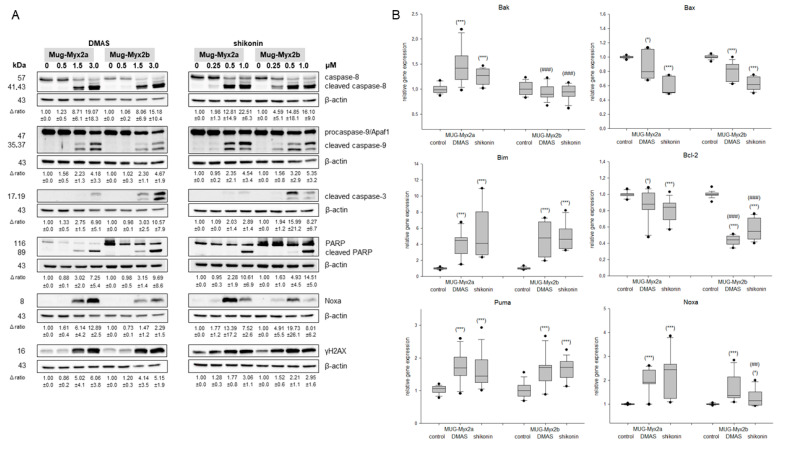
Apoptotic induction. (**A**) Protein expression of cleaved caspase-8, -9, and -3, as well as cleaved PARP, Noxa, and the DNA damage marker γH2AX. The apoptotic key players were evaluated by immunoblotting under control conditions (0) and after treatment with 0.5, 1.5, and 3 µM DMAS respectively 0.25, 0.5, and 1.0 µM shikonin. β-actin was used as loading control. Δ ratio, fold change was normalized to non-treated controls (mean ± SD of *n* = 3). Full-length blots are presented in Appendix A. (**B**) Relative gene expression of the pro-apoptotic markers *Bak, Bax* and *Bim*, the anti-apoptotic marker *Bcl-2*, and the antagonists *Puma* and *Noxa* after treatment with DMAS or shikonin for 24 h in MUG-Myx2a and MUG-Myx2b cells (mean ± SD, *n* = 6, measured in triplicate). Statistical significances to the untreated controls are defined as follows: * *p* < 0.05; *** *p* < 0.001. Statistical significances between the two cell lines MUG-Myx2a and MUG-Myx2b are presented as ^##^
*p* < 0.01; ^###^
*p* < 0.001.

**Figure 3 ijms-24-15910-f003:**
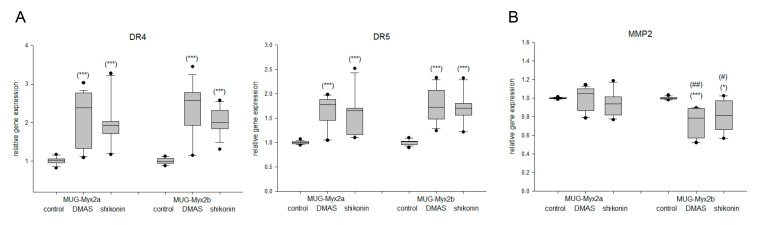
Altered gene expression of death receptors and MMP2. (**A**) Relative gene expression analysis of DR4 (*TNFRSF10A*), DR5 (*TNFRSF10B*), and (**B**) the metastasis factor *MMP2* after treatment with DMAS and shikonin for 24 h in MFS cells. Untreated control cells served as reference value (ratio = 1; mean ± SD, *n* = 6, measured in triplicate). Statistical significances to the untreated controls are defined as follows: * *p* < 0.05; *** *p* < 0.001. Statistical significances between the two cell lines MUG-Myx2a and MUG-Myx2b were marked by means of ^#^
*p* < 0.05; ^##^
*p* < 0.01.

**Figure 4 ijms-24-15910-f004:**
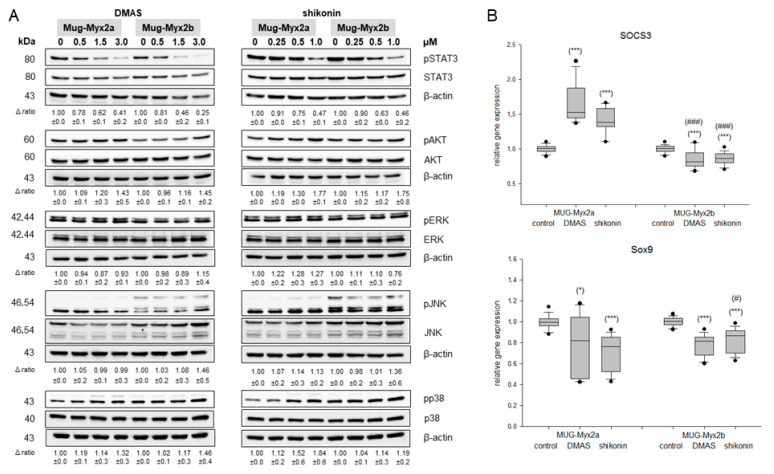
MAPK phosphorylation. (**A**) Protein phosphorylation of STAT3, AKT, ERK, JNK and p38 was analyzed by immunoblotting under control conditions (0) and after treatment with 0.5, 1.5, and 3 µM DMAS, respectively 0.25, 0.5, and 1.0 µM shikonin. β-actin was used as loading control. Δ ratio, fold change normalized to non-treated controls (mean ± SD of *n* = 3). Full-length blots are presented in Appendix A. (**B**) Relative gene expression of the STAT3 downstream targets *SOCS3* and *Sox9* after treatment with DMAS or shikonin for 24 h in MUG-Myx2a and MUG-Myx2b cells (mean ± SD, *n* = 6, measured in triplicate). Statistical significances to the untreated controls are defined as follows: * *p* < 0.05; *** *p* < 0.001. Statistical significances between the two cell lines MUG-Myx2a and MUG-Myx2b are presented as ^#^
*p* < 0.05; ^###^
*p* < 0.001.

**Figure 5 ijms-24-15910-f005:**
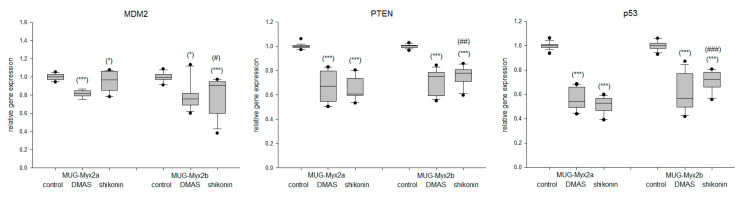
Altered gene expression of the *MDM2, PTEN*, and *p53* tumor-suppressor-oncoprotein network. Relative gene expression analysis of *MDM2, PTEN*, and *p53* after treatment with DMAS and shikonin for 24 h in MFS cells. Untreated control cells served as reference value (ratio = 1; mean ± SD, *n* = 6, measured in triplicate). Statistical significances to the untreated controls are defined as follows: * *p* < 0.05; *** *p* < 0.001. Statistical significances between the two cell lines MUG-Myx2a and MUG-Myx2b were marked by means of ^#^
*p* < 0.05; ^##^
*p* < 0.01; ^###^
*p* < 0.001.

**Figure 6 ijms-24-15910-f006:**
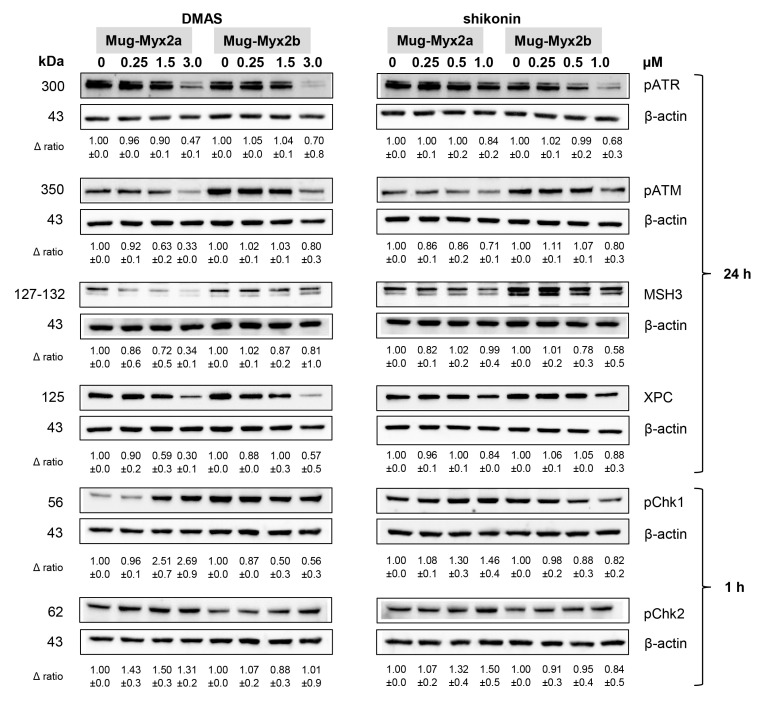
DNA damage response after DMAS and shikonin treatment. Protein expression and phosphorylation of DNA damage markers pATR, pATM, MSH3, and XPC, as well as pChk1 and pChk2 was analyzed by immunoblotting under control conditions (0) and after treatment with 0.5, 1.5, and 3 µM DMAS, respectively 0.25, 0.5, and 1.0 µM shikonin. β-actin was used as loading control. Δ ratio, fold change normalized to non-treated controls (mean ± SD of *n* = 3). Full-length blots are presented in Appendix A.

**Table 1 ijms-24-15910-t001:** Detailed information on the used antibodies. Abbreviations: “p” means phosphorylated; “ON” means “over night”.

**Target**	**Company;** **Order Number**	**Host Species**	**Clonality**	**Dilution**	**Incubation Time**
cCaspase 8	Santa Cruz; Sc-81656	mouse	monoclonal	1:1000	ON 4 °C
cCaspase 9	Santa Cruz; Sc-56076	mouse	monoclonal	1:1000	ON 4 °C
cCaspase 3	Cell signaling; CS, 9661	rabbit	polyclonal	1:1000	ON 4 °C
cPARP	ThermoFisher; 436400	mouse	monoclonal	1:1000	ON 4 °C
Noxa	ThermoFisher; MA1-41000	mouse	monoclonal	1:1000	ON 4 °C
STAT3	Cell signaling; CS, 4904	rabbit	monoclonal	1:1000	ON 4 °C
pSTAT3	Cell signaling; CS, 9145	rabbit	monoclonal	1:1000	ON 4 °C
AKT	Cell signaling; CS, 9272	rabbit	polyclonal	1:1000	ON 4 °C
pAKT	Cell signaling; CS, 9271	rabbit	polyclonal	1:1000	ON 4 °C
ERK	Cell signaling; CS, 4695	rabbit	monoclonal	1:1000	ON 4 °C
pERK	Cell signaling; CS, 4370	rabbit	monoclonal	1:1000	ON 4 °C
JNK	Cell signaling; CS, 9252	rabbit	polyclonal	1:1000	ON 4 °C
pJNK	Cell signaling; CS, 9251	rabbit	polyclonal	1:1000	ON 4 °C
p38	Cell signaling; CS, 9212	rabbit	polyclonal	1:1000	ON 4 °C
pp38	Cell signaling; CS, 4511	rabbit	monoclonal	1:1000	ON 4 °C
pATR	Cell signaling; CS, 2853	rabbit	polyclonal	1:1000	ON 4 °C
pATM	Cell signaling; CS, 13050	rabbit	monoclonal	1:1000	ON 4 °C
MSH3	Santa Cruz; Sc-271080	mouse	monoclonal	1:1000	ON 4 °C
XPC	Santa Cruz; Sc-74410	mouse	monoclonal	1:1000	ON 4 °C
pChk1	Cell signaling; CS, 2348	rabbit	monoclonal	1:1000	ON 4 °C
pCHk2	Cell signaling; CS, 2197	rabbit	monoclonal	1:1000	ON 4 °C
β-actin	Santa Cruz; Sc- 47778	mouse	monoclonal	1:15,000	ON 4 °C

## Data Availability

Not applicable.

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
