# Peer review of "The Biological Assessment of Shikonin and β,β-dimethylacrylshikonin Using a Cellular Myxofibrosarcoma Tumor Heterogeneity Model"

_ijms, 2023, doi:10.3390/ijms242115910_

Round 1

Reviewer 1 Report

Comments and Suggestions for Authors

A comprehensive study showing the potential for using shikonin and its derivative for treatment of myxofibrosarcoma, with extensive analysis of the mechanism of action in vitro.

My main issue is the lack of a standard clinical agent which has demonstrated similarly good in vitro activity versus myxofibrosarcoma as a comparative control in the studies. What is to say that similar deficiencies which are seen for standard agents in the clinic for this cancer will not be seen with the compounds described here?

I think a more convincing case would be put by demonstrating improved efficacy vs the control agents in these models. Additionally more extensive studies exploring the pharmacokinetics and pharmacodynamics of the compounds in vivo should be considered.

Comments on the Quality of English Language

Quality is on the whole good, with just some minor typographical and grammatical errors

Reviewer 2 Report

Comments and Suggestions for Authors

1) In Figure 1B, because shikonin or DMAS is poorly soluble in water, it required to show the effect of solvent on the cell viability and proliferation as a control in the main figure. 

2) In figure 1B, because authors specified the effective reduction in cell vialbilty in both cancer cell lines (MUG-Myx2a & MUG-Myx2b), it is required to demonstrate the effects of DMAS and shikonin and in a non-tumorous primary cell lines (which most closely represent the tissue of origin) as a control experiment.  

3) Apparently, in most of the figure 2, the loading controls (beta-Actin) bands don't look even, then how the ratio has been calculated for the upregulation ? Please justify those fold changes with respect to those loading controls. Also, please check all the western blots carefully for the same reason. 

Comments on the Quality of English Language

The quality of English is very good. 

Reviewer 3 Report

Comments and Suggestions for Authors

Dear Authors,

This manuscript is interesting and important in the context of anticancer drug development. Only two methods were used in the study: western blot and PCR and this is the main deficiency of this work. However, these methods were used extensively and well planned. Some parts of the manuscript and details need to be corrected before publication.

1. statistical analysis; it is the lack of information of normality distribution of variables, it must be added. Also, Student's t-test or Wilcoxon test were used; these tests are only useful for comparing two groups; in this study more groups were analysed and other tests should be used, such as Kruskal-Wallis (for non-parametric non-normal distribution of variables). 

2. It is the lack of producer names (cell culture section) or city and country in some parts of the Materials and Methods.

Materials and methods need to be more detailed in the description for recapitulation of the study. 

3. Western blot: please include antibody dilutions, incubation times, catalog numbers, host species and clonality. Reproduction of these results is not possible in this state.

4. PCR methods: please provide catalog numbers of primers and manufacturers.

5. Please include the name of the producer and the city and country if the producer's name is used for the first time in any manuscript. 

6.  Introduction: There is not enough information of myxofibrosarcoma about epidemiology, mortality, targets in this therapy. There is a need to justify the provision of research in this area. Correction of this section will increase the value of this publication.

Round 2

Reviewer 1 Report

Comments and Suggestions for Authors

The authors have addressed my concerns, although I think it would be helpful to give a little more detail of the failure of doxorubicin in the introduction and discussion, and as to why the approach taken here should be more successful.

Author Response

Authors reply: Thank you for your positive feedback on our revision. We have added the topic of doxorubicin in the Discussion in the following sentences:

Adjuvant chemotherapy consists, as in most sarcoma entities, of first line therapy with the anthracycline doxorubicin. However, it is not well suited as a comparative substance in cell culture, because it is known that authentic doxorubicin is converted in cell culture media to a chemically distinct form resulting in varying chemosensitivity and activity. This transformation results in a significant loss of lethality in vitro while retaining antiproliferative activity [17].” (lines 262-267)

Ref. #17: Pavlik, E.J.; Kenady, D.E.; van Nagell, J,R, Jr; Hanson, M.B.; Donaldson, E.S.; Casper, S.; Garrett, D.; Smith, D.; Keaton, K.; Flanigan, R.C. Stability of doxorubicin in relation to chemosensitivity determinations: loss of lethality and retention of antiproliferative activity. Cancer Invest 1984, 2(6), 449-458.

Thank you for your time and efforts to improve our publication.

Reviewer 2 Report

Comments and Suggestions for Authors

Dear Editor/s

The authors of the research paper "The biological assessment of shikonin and β,β-dimethylacrylshikonin using a cellular myxofibrosarcoma tumor heterogeneity model" has responded and provided adequate infomation and supportive data from my previous comments/concerns. Therefore, I would recomment this for futher acceptance process of the paper. 

Thank you 

Author Response

Authors reply: I am pleased that we were able to satisfactorily process all suggestions and thank you for your endorsement of the publication.

Reviewer 3 Report

Comments and Suggestions for Authors

Dear Authors,

all corrections have greatly improved the quality of this manuscript. I recommend this interesting manuscript for publication.

Author Response

(The authors gave the same response as above.)
